# Association of Sedentary Behavior and Depression among College Students Majoring in Design

**DOI:** 10.3390/ijerph17103545

**Published:** 2020-05-19

**Authors:** Zhen Xu, Qiuxia Xu, Yifan Wang, Jielu Zhang, Jiapei Liu, Fei Xu

**Affiliations:** 1College of Landscape Architecture, Nanjing Forestry University, Nanjing 210037, China; xuqiuxia@njfu.edu.cn (Q.X.); mywj87630@163.com (Y.W.); zhangjielu@njfu.edu.cn (J.Z.); Liujiapei1105@163.com (J.L.); 2Nanjing Municipal Center for Disease Control and Prevention, Nanjing 210003, China; frankxufei@163.com

**Keywords:** sedentary behavior, depression, college student, occupational health, design students

## Abstract

Background: This study aimed to specify the prevalence of sedentary behavior and depression and investigate the relationship between sedentary behavior and depression among college students majoring in design. Methods: A total of 480 undergraduate and postgraduate students majoring in design were randomly enrolled from a university in Nanjing for a questionnaire that included sociodemographic data, physical health, sedentary behavior and depression. Results: Participants reported that they spent 14.93 (SD = 1.76) hours on sedentary behavior per day and most of the time occurred outside the classroom. There were 161 (39.8%) students who reported depression, with a statistical difference across grades. After adjusting for sociodemographic attributes, physical health and physical activity, binary logistic regression analysis showed that the total sedentary time and time spent on school assignments on weekends were significantly associated with depression. Conclusions: To reduce the risk of depression, students majoring in design should be encouraged to change sedentary behaviors to physical activities in their study and life, such as using non-seating postures to do school assignments, making time for more physical activities and reducing assignments on weekends.

## 1. Introduction

Sedentary behavior is any waking behavior characterized by an energy expenditure ≤1.5 METs while in a sitting or reclining posture [1,2], including watching TV, reading, working on a computer or lying down with a phone. Its prevalence has increased in most countries and regions of the world in recent decades, due to the change of lifestyle and the soaring demand for brainworkers [3,4]. According to related data from the World Health Organization (WHO), one-quarter of adults have insufficient physical time and three-quarters of adults have too much sedentary time [5]. Sedentary behavior also places a severe burden on the health care system. In America, the medical expenses caused by sedentary behavior among urban residents have reached $150 billion a year and have been on the rise for nearly a decade [6]. A sedentary lifestyle could become one of the major public health problems in this century.

Recently published evidence suggests that the prevalence of sedentary behaviors is negatively associated with health outcomes such as obesity, cardiovascular illnesses and metabolic disorders [7]. Following 50,277 non-obese women for five years, Hu et al. provided evidence for the link between sedentary behavior and health, showing that an increase of two hours of TV per day was associated with a 23% increase in obesity [8]. As a major mental health disorder, depression has threatened more than 350 million people worldwide [9]. However, the relationship between sedentary behavior and mental health has not received enough attention, especially in the absence of relevant studies in other domains (e.g., work related and entertainment related).

Designers are at a greater risk of depression and suicide due to their notorious amount of working hours. Furthermore, these population groups are also at increased risk of a high amount of sedentary behavior in front of a computer. The behavioral habits of most adults, such as physical activity, sedentary behavior and lifestyle, are considerably affected by habits in college [10]. Relevant research showed that if physical activity was sufficient during college, 85% of people still had adequate physical activity in the 6 years after graduation, while 82% of students with insufficient physical activity during college remained inactive for a long time [10]. Therefore, university life is the key period to determine the adult health lifestyle. However, there is tremendously insufficient research on the lifestyle and mental health of designers and their pre-career students.

The purpose of the current study was to investigate the prevalence of sedentary behavior (e.g., study and entertainment) and risk of depression using data from a sample of college students majoring in design, aiming to provide references for the active intervention of mental health in the built environment and management system.

## 2. Materials and Methods

### 2.1. Methods

The study invited 500 design students to participate, from full-time undergraduate and graduate students in Nanjing Forestry University, of which 480 (96%) agreed to participate. The criteria for inclusion in the study were students who majored in landscape architecture, urban and rural planning, visual communication, environmental design, environmental art design or public art.

This was a cross-sectional study conducted in November 2017, in the middle of the semester, to purposely avoid seasonal holidays and examination time. The data were collected using a questionnaire. To ensure consistency, all questionnaires were administered face-to-face using printed paper under the direction of investigators. With the written consent of the respondents, questionnaires were distributed and collected in their dormitories by trained investigators within 2 weeks. The questionnaire was divided into four sections: basic sociodemographic attributes (gender, grade, relationship status, living expenses); physical health (height, weight, frequency of cold, 800-m test score), sedentary behavior and physical activity (7-day activities recall); and depression.

### 2.2. Measures

#### 2.2.1. Sedentary Behavior and Physical Activity

It is difficult to accurately evaluate sedentary behavior through a questionnaire because it takes place in various sporadic forms in daily life, which makes subjective measurement more complicated [11,12]. Sedentary behavior questionnaire use is a developing field, and there is not a consensus hitherto on a reliable questionnaire for subjective measurement to quantify sedentary activity, especially occupation [11]. However, people can report physical activities that can be remembered and described more accurately (e.g., playing football, riding a bike, etc.) than trivial and various types of sedentary behavior (e.g., watching TV, reading, driving, playing games, lying in a chair, etc.). The movement behaviors are divided into physical activity, sedentary behavior and sleep [1]. Therefore, the total sedentary time per day in this study is in addition to time spent on sleeping and physical activity in 24 h. Physical activity time was obtained from the International Physical Activity Questionnaire (IPAQ) short form report [13]. Meanwhile, respondents also reported the time of getting up and the time of falling asleep. Moreover, the relationships between different types of sedentary behavior and depression are different, and it is necessary to analyze subdivided sedentary behaviors in order to make specific suggestions for the sedentary behavior of design students. These can be divided into necessary sedentary behavior (class activities), which can be calculated based on class schedules, and optional sedentary behavior (school assignments, extracurricular learning, entertainment). Participants were asked to report hours they spent per day over the past seven days doing school assignments, learning extracurricular knowledge and using entertainment, separately for weekdays and weekends. Each item was scored on a five-point scale, with the following response options: <1 h, 1–3 h, 3–6 h, 6–10 h and over 10 h. Using this format, the screen-based sedentary activity of design students on weekdays and weekends was included in this questionnaire.

#### 2.2.2. Sociodemographic Attributes

Due to the differences in sedentary behavior between males and females, data were stratified by gender (men = 1, women = 2), resulting in samples of 317 females and 163 males. Participants self-reported their grade (first-year students, second-year students, third-year students, fourth-year students, first-year graduate students, second-year graduate students, third-year graduate students). The design students’ relationship status was also included, reporting single or not. To assess socioeconomic status (SES), monthly living expenses were reported with a five-point scale (0–500 yuan, 500–1000 yuan, 1000–1500 yuan, 1500–2000 yuan, 2000 yuan and above).

#### 2.2.3. Physical Health

The measure of physical health included body mass index (BMI), frequency of cold and 800-m test score. Height and weight were reported to calculate the BMI score as an approximate index to measure individual obesity [14]. BMI was calculated as the weight in kilogram divided by the square of height in meters, and categorized to four levels: underweight, normal, overweight and obese. Students also reported on the frequency of having a cold in a year on a five-point scale from “none or 1 time” to “more than 6 times”. Finally, their 800-m test score was included with three-point scale (below average, average, and above average). This question was developed by the research team according to the physical fitness tests of students in China.

#### 2.2.4. Depression

The Zung self-rating depression scale (SDS) was used to evaluate depression of design students [15]. In the SDS, respondents self-reported how often they had experienced symptoms of 20 conditions during the preceding week on a 4-point ordinal response scale from “a little of the time” to “most of the time”. The total score of the SDS was converted into a standard score, and a score ≥50 was classified as positive.

### 2.3. Statistical Analysis

All statistical analyses were performed using IBM SPSS (version 22.0 IBM Corp, Armonk, NY, USA). Continuous data were checked for normality and variables were presented as mean (±standard deviation; SD) and median (interquartile range; IQR). Categorical variables were described using frequencies and percentages. The differences between different groups of individual attributes with or without depression were assessed using a chi-square test. For the multivariate analysis, the main analysis consisted of logistic regression analysis using the binary depression (i.e., depression or not depression) as the outcome. Odds ratios (ORs) and b-coefficients with 95% confidence intervals (CIs) were calculated from the regression analyses. Potential confounders included in the multivariate models were basic sociodemographic attributes, physical health and physical activity levels. The significance level was set at *p* < 0.05.

## 3. Results

### 3.1. Participant Characteristics

A total of 480 design students (317 women and 163 men) were included in the analysis. The majority of respondents were undergraduate students majoring in design (85%); 15% of the sample were postgraduate students. 21.7% were first-year students, 17.7% were second-year students, 23.3% were third-year students, 22.3% were fourth-year students, and the rest were postgraduate students.

### 3.2. Prevalence of Sedentary Behavior

In the week preceding the questionnaire, respondents spent 14.93 (SD = 1.76) hours every day on sedentary behavior, in which 4.14 h a day was spent on class time and 10.79 h a day was spent on optional sedentary behavior. The proportions of time reported optional sedentary behavior in each domain are shown in Table 1 on weekdays and weekends respectively. More than half of respondents (51.8%) reported spending more than 3 h a day on school assignments on weekends and 9.3% of respondents reported spending more than 10 h a day on school assignments on weekends. The total sedentary time of male and female students was 14.76 (SD = 1.76) hours a day and 15.04 (SD = 1.76) hours a day respectively with no significant difference. However, participants might show more sedentary behavior if they are sophomore design students.

The majority of the respondents reported screen time of over 3 h a day on both weekdays and weekends, and only a few reported screen times of less than 1 h a day. The results are shown in Table 2. There was no significant difference in the time per day spent using a screen between males and females or different grades.

### 3.3. Individual Attributes and Depression

In this survey, 39.8% of respondents were depressive. Among them, 1.5% showed intense depressive symptoms, while 15.8% reported mild-to-severe depressive symptoms and the rest reported mild-to-moderate depressive symptoms. Table 3 shows various sociodemographic and lifestyle characteristics of the participants who expressed depression. There was no statistical difference between the depression and no depression groups except grade and frequency of cold.

There is a relationship between various types of sedentary behavior and there were separate regressions for depression or non-depression. In models (Table 4) adjusted for gender, grades, living expenses, relationship status, cold frequency, 800-m test performance, obesity and physical activity, more total sedentary behavior (above average) was associated with a 4.31 (95% CI = 1.22–15.15) times higher odds for depression, and more school assignments with sedentary behavior on weekends was also associated with a 3.56 (95% CI = 1.35–9.38) times higher odds for depression. However, other sedentary behaviors were not associated with depression.

## 4. Discussion

This study shows that the individual sedentary time of students majoring in design in Nanjing Forestry University is 14.93 (SD = 1.76) hours a day. In the United States, the United Kingdom, Canada and other developed countries, studies have found that the sitting time of adults is 7.3–11.2 h a day [16,17,18,19]. Due to the particularity of the design profession, the severe problem of sitting too long among design students should receive more attention. The screen time of design majors is longer than that of other related research [20], which may be attributed to relying on a computer to complete most of their homework. There is a significant difference in sedentary time between different grades of students. Due to the beginning of the design skills training and the significant increase in design courses, which requires more time for design exercises, a sophomore’s sedentary time is usually longer.

The study of factors related to depression helps to identify possible risk factors and strategies to prevent depression. The prevalence of depression is 39.8%, which is higher than that found by Cui et al. [21] and He et al. [22]. This may be caused by a large number of school assignments among students majoring in design, the diverse range of design assignments, which often include a deadline, and the rather severe employment situation of the design industry. In this study, the proportion of males reporting depression is higher than that of females, although no significant difference was found. Our study has shown that different grades have different risks of depression. Generally speaking, due to writing a thesis and employment preparation, the psychological pressure of graduate students may be greater. The small sample of graduate students in this survey also reflects that the risk of depression is higher than other grades. 

When adjusting for socioeconomic attributes, physical health and physical activity, the results of this study suggested that students who spent a long time on school assignments during the weekend and on total sedentary behavior were more likely to be depressive. This is consistent with the European and American literature and confirms observations that depression symptoms of students majoring in design are associated with total sedentary time [16,23]. Vallance et al. conducted a National Health and Nutrition Examination Survey of 2862 American adults and found that people with long periods of sedentary behavior were more likely to suffer depression [23]. Moreover, a meta-analysis by Zhai et al. proved that the relationship between sedentary time and depression was independent of the study area, study design, type of sedentary activity, type of depression and whether it controlled for physical activity [24]. For design students, more time spent on sedentary behavior may reduce physical ability, reduce the opportunities for communication with classmates and lower students’ social needs, resulting in depression. However, due to the cross-sectional study design in this article, the mechanism of the relationship between sedentary time and depression is currently unclear. This may be due to the fact that students with depression tend to participate in more sedentary activities, or too much time spent sitting may cause depression, or there are other intermediate variables. Reviews of previous research can provide some indication to underpin the mechanisms underlying the effect of sedentary time on depression. A sedentary intervention study suggested that negative mood or stress following experimentally induced sedentary time was associated with pro-inflammatory symptoms [25]. Besides, social interaction, which is related to social anxiety and depression, may explain this mechanism as prolonging sedentary behavior may lead to social solitude and removal from social support [26]. Furthermore, as the screen-based sedentary behavior accounts for a large proportion of total sedentary time, these activities could also increase the likelihood of sleep and mood disorders [27]. Except for time spent on school assignments on weekends, the composition of various types of sedentary behavior does not affect depression of design students. At present, there are few studies on the relationship between school assignment time on weekends and depression. Regarding the types of sedentary behavior, only design students who spend a long time on school assignments during weekends were more likely to increase their risk of depression, which may be due to the pressure of assignments and poor time management.

We suggest that students can be encouraged to shift some of their sedentary behavior to physical activities in terms of teaching, management and physical environment. For example, in the process of design training and coursework, the instructor could enrich the form of assignments by adding assignments that can be completed without sitting. Schools can encourage students to manage their physical activities and reduce time spent on assignments during weekends. Regarding the built environment of the university, regular, long-term sedentary behavior can be partially reduced by establishing a non-seated learning space [28]. The outdoor environment of the campus should be redesigned to promote leisure physical activities of students [2], thereby reducing sedentary time.

There are some limitations to the present study that should be addressed. This cross-sectional study cannot suggest evidence for causality, but future research may confirm these findings. In this study, the questionnaire participants are design students of one university, and the results cannot be compared across universities. Furthermore, the research results were obtained from the subjective reports of participants rather than objective measurements. Participants may report more sedentary behaviors than actual behaviors, which may be due to less physical activity and walking time reported by IPAQ [29], or the physical activity reported by IPAQ does not include walking and standing activities lasting for less than 10 min. However, using the questionnaire provided a large sample size for our research at low cost. A combination of accelerators and questionnaires can be used to ensure the number of participants and the accuracy of sedentary behavior in further research.

## 5. Conclusions

This study provides evidence of the association between depression and total sedentary time and extends previous findings by showing that there is no relationship between subdivided sedentary time and depression, except for school assignment hours on the weekend. This association persisted independently of differences in gender, grade, living expenses, single, cold frequency, 800-m test result, obesity and physical activity. Our results support the active lifestyle that guides designing students towards more physical activity and less sedentary/school assignment time. Further research on directional/causal relationships between total sedentary time, school assignment hours and depression are necessary for more precisely positive interventions.

## Figures and Tables

**Table 1 ijerph-17-03545-t001:** Optional sedentary behavior of students majoring in design on workdays and weekends across each domain.

		<1 h	1–3 h	3–6 h	6–10 h	10 h and Above
Weekdays	School assignment	51 (10.9)	161 (34.6)	126 (27.0)	79 (17.0)	49 (10.5)
	Extra-curricular learning	127 (27.2)	203 (43.5)	78 (16.7)	35 (7.5)	24 (5.1)
	Entertainment	24 (5.1)	223 (47.8)	155 (33.2)	32 (6.8)	33 (7.1)
Weekends	School assignment	61 (13.2)	162 (35.0)	138 (29.8)	59 (12.7)	43 (9.3)
	Extra-curricular learning	119 (25.4)	215 (45.9)	78 (16.6)	39 (8.3)	18 (3.8)
	Entertainment	28 (6.1)	175 (37.9)	194 (42.0)	46 (9.9)	19 (4.1)

**Table 2 ijerph-17-03545-t002:** Screen time of students majoring in design on weekdays and weekends.

	<1 h	1–3 h	3–6 h	6–10 h	10 h and Above
Weekdays	31 (6.7)	197 (42.8)	145 (31.5)	44 (9.5)	44 (9.5)
Weekends	18 (3.9)	173 (37.6)	165 (35.9)	68 (14.8)	36 (7.8)

**Table 3 ijerph-17-03545-t003:** Sociodemographic and physical health characteristics of participants.

	Depression	No Depression	*p*
Gender			0.50
Men	58 (42.0)	80 (58.0)	
Women	103 (38.6)	164 (61.4)	
Grade			<0.01 *
First-year students	23 (26.4)	64 (73.6)	
Second-year students	34 (54.0)	29 (46)	
Third-year students	52 (53.1)	46 (46.9)	
Fourth-year students	27 (30.3)	62 (69.7)	
First-year graduate students	6 (20.0)	24 (80.0)	
Second-year graduate students	16 (48.5)	17 (51.5)	
Third-year graduate students	3 (60.0)	2 (40.0)	
Living expenses			0.34
0–500 yuan	6 (66.7)	3 (33.3)	
500–1000 yuan	25 (45.5)	30 (54.5)	
1000–1500 yuan	58 (40.3)	86 (59.7)	
1500–2000 yuan	41 (35.0)	76 (65.0)	
2000 yuan and above	25 (39.7)	38 (60.3)	
Single			0.23
Yes	109 (41.1)	156 (58.9)	
No	38 (34.5)	72 (65.5)	
Frequency of cold			<0.01 *
None or 1 time	103 (70.1)	44 (29.9)	
2–3 times	111 (61.7)	69 (38.3)	
3–6 times	28 (75.7)	9 (24.3)	
More than 6 times	10 (76.9)	3 (23.1)	
800-m test result			0.12
Good	28 (49.1)	29 (50.9)	
Average	76 (42.0)	105 (58.0)	
Bad	34 (33.0)	69 (67.0)	
Overweight or obese			0.45
Yes	11 (33.3)	22 (66.7)	
No	103 (40.2)	153 (59.8)	

* *p* for statistical difference between participants with and without depression.

**Table 4 ijerph-17-03545-t004:** Binary logistic regression for depression by total sedentary behavior and school assignments of sedentary among design students.

	B	SE	Wald	P	OR (95% CI)
Total sedentary time					
Below average					Ref.
Above average	1.46	0.64	5.18	0.02	4.308 (1.22–15.15)
School assignments					
<3 h					Ref.
>3 h	−1.27	0.50	6.59	0.01	3.56 (1.35–9.38)

Adjusted for gender, grade, living expenses, relationship status, cold frequency, 800-m test result, obesity and physical activity. OR: odds ratio; CI: confidence interval.

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
