# Peer review of "Association of Sedentary Behavior and Depression among College Students Majoring in Design"

_ijerph, 2020, doi:10.3390/ijerph17103545_

Round 1

Reviewer 1 Report

Although this actual study was done in 2017 it is very crucial today in light of the pandemic that is effecting a number of students being more sedentary and increase in anxiety and depression. The strength of the study is the number subjects. The limitation as stated by the researchers is the subjectivity of the questionnaire. If the study were to be replicated as stated in the conclusion was for future measures. Results of the study had sound statistics and breakdown of quality and quantity of students spent engaged in physical activity and sedentary. the breakdown of male, femaies and year of student was also significant in the study. Another strength of the study is relating to studies in America and Europe for relations to other cultures.

Author Response

Dear reviewer:

Thank you for acknowledging the importance of this work. Your comments are pertinent and insightful for revising and improving our manuscript. The revision, relevant to your review, has been highlighted in yellow in the manuscript. Our responses are as following,

Comment: The limitation as stated by the researchers is the subjectivity of the questionnaire. If the study were to be replicated as stated in the conclusion was for future measures.

Response: For measurement of sedentary behaviour, it’s beneficial to expand the survey methods. Therefore, in the limitations section, we proposed in the further research that the accelerometer and the questionnaire can be combined to obtain the participants' sedentary time more credibly and accurately while at a certain cost.

Since this survey was completed in 2017, some students have graduated. So, it is impossible now to add objective-measured data of sedentary activity. Actually, studies of college students especially of design major has not received attention in the authors region, for which our Chinese manuscript of this study was rejected just for “college student is not an occupation” and “should not focus only on students of one major”, etc. So far, we still have no sufficient support for three-axis accelerators to measure physical activity precisely.

However, in our study, the total sedentary time was obtained by the international physical activity questionnaire (IPAQ) and sleeping time. IPAQ has been tested for reliability and validity for various ages professionally and internationally. For the interviewee, the time to fall asleep and wake-up was quite regular and easy to recall. Therefore, the sedentary activities here obtained from the questionnaire should be reliable, as many other published papers proved.

To respond to another reviewer’s comments, we have added the details of the manuscript and the interpretation of the results. we re-proofread the manuscript and found typos on lines 149 and 150. This did not lead to changes in results and conclusions and we apologized for this mistake.

Reviewer 2 Report

The topic is interesting and the research design is sound. However, I would suggest to provide more details regarding the used questionnaires (i.e. self-report, number of items, sample items). In addition, the discussion section should be improved, expanding on the possible interpretation of the results. For example, what about the link between inflammation/oxidative stress and depression in relation to sedentary lifestyle?

Author Response

Dear Reviewer,

Thank you for your valuable and very helpful comments. We have revised accordingly after reading your opinions carefully. The revision, relevant to your review, has been highlighted in green in the manuscript. Our responses are as following,

1.Comment: I would suggest to provide more details regarding the used questionnaires (i.e. self-report, number of items, sample items).

Response: Flowing your suggestion for the details of questionnaires, in the revised version, we have added the details of self-report, the number of items and sample items to present the content of the questionnaire more clearly.

2.Comment: The discussion section should be improved, expanding on the possible interpretation of the results. For example, what about the link between inflammation/oxidative stress and depression in relation to sedentary lifestyle?

Response: Thanks for pointing out the insufficient interpretation of the results. After reviewing some relevant literature, we have added the possible interpretation for the relationship between sedentary behavior and depression, inflammation, sleep disorders, social isolation.

We re-proofread the manuscript and found typos on lines 149 and 150. This did not lead to changes in results and conclusions and we apologized for this mistake.
